# In Vitro and In Vivo Biocompatibility Studies on Engineered Fabric with Graphene Nanoplatelets

**DOI:** 10.3390/nano12091405

**Published:** 2022-04-20

**Authors:** Carla Fanizza, Mara Stefanelli, Anna Risuglia, Erika Bruni, Federica Ietto, Federica Incoronato, Fabrizio Marra, Adele Preziosi, Patrizia Mancini, Maria Sabrina Sarto, Daniela Uccelletti

**Affiliations:** 1Department of Technological Innovations and Safety of Plants, Products and Anthropic Settlements (DITSIPIA), National Institute for Insurance against Accidents at Work (INAIL), 00143 Rome, Italy; m.stefanelli@inail.it (M.S.); a.risuglia@inail.it (A.R.); f.ietto@inail.it (F.I.); f.incoronato@inail.it (F.I.); 2Department of Biology and Biotechnology C. Darwin, Sapienza University of Rome, 00185 Rome, Italy; erika.bruni@uniroma1.it (E.B.); adele.preziosi@uniroma1.it (A.P.); daniela.uccelletti@uniroma1.it (D.U.); 3Department of Astronautical, Electrical and Energy Engineering, Sapienza University of Rome, 00184 Rome, Italy; fabrizio.marra@uniroma1.it (F.M.); mariasabrina.sarto@uniroma1.it (M.S.S.); 4Research Center for Nanotechnology Applied to Engineering, Sapienza University of Rome, 00184 Rome, Italy; 5Department of Experimental Medicine, Sapienza University of Rome, 00161 Rome, Italy; patrizia.mancini@uniroma1.it

**Keywords:** smart fabrics, strain sensors, graphene nanoplatelets, biocompatibility, HaCaT cells, MTT assay, FESEM analysis, immunofluorescence microscopy analysis, *Caenorhabditis elegans*

## Abstract

To produce clothes made with engineered fabrics to monitor the physiological parameters of workers, strain sensors were produced by depositing two different types of water-based inks (P1 and P2) suitably mixed with graphene nanoplatelets (GNPs) on a fabric. We evaluated the biocompatibility of fabrics with GNPs (GNP fabric) through in vitro and in vivo assays. We investigated the effects induced on human keratinocytes by the eluates extracted from GNP fabrics by the contact of GNP fabrics with cells and by seeding keratinocytes directly onto the GNP fabrics using a cell viability test and morphological analysis. Moreover, we evaluated in vivo possible adverse effects of the GNPs using the model system *Caenorhabditis elegans*. Cell viability assay, morphological analysis and *Caenorhabditis elegans* tests performed on smart fabric treated with P2 (P2GNP fabric) did not show significant differences when compared with their respective control samples. Instead, a reduction in cell viability and changes in the membrane microvilli structure were found in cells incubated with smart fabric treated with P1. The results were helpful in determining the non-toxic properties of the P2GNP fabric. In the future, therefore, graphene-based ink integrated into elastic fabric will be developed for piezoresistive sensors.

## 1. Introduction

Within the last few years, wearable technology has extensively developed with hundreds of products such as smartwatches, smart clothes and smart tools for medical uses.

Wearable systems capable of estimating multiple physiological parameters (ECG, respiratory rate, body temperature, etc.) are used in various sectors such as fitness, health, information, entertainment, industrial sector, military sector, family assistance and workplaces.

In the smart clothes field, smart fabrics have been developed due to the combination of engineering, electronics and nanotechnology aiming to create nanotextiles, common textiles incorporating nanoparticles, which can be involved in many different fields of application, exploiting their electrical, conductive, optical or antibacterial properties [1,2].

Moreover, ergonomics, durability (abrasion resistance and washability) and biocompatibility are also important key factors that need to be considered in the development of functional and smart apparel products made with smart fabric.

In addition to the design, development and functionality, a particular interest in smart clothes is the comfort of the end-user [3,4,5,6].

In the literature, the production of various wearable devices that are based on nanomaterials is reported. For example, Zhang et al. show commercial fabrics coated with conductive polymer bath [7]; Ilanchezhyan et al. present cotton fibers coated with functionalized Carbon Nanotubes (CNTs) to realize wearable heating devices [8].

For monitoring human motion (thumb movement, walking, running, etc.) [9], wearable sensors have been developed with silver nanowires used as electrodes and Ecoflex as a dielectric. Sensitive piezoelectric sensors fabricated with PVDF (polyvinylidenefluoride-co-trifluoroethylene) nanofiber mat are exploited for monitoring radial and carotid pulses [10].

Y Ai et al. report a multifunction sensor manufactured with reduced graphene oxide (rGO) encapsulated Poly(vinylidene fluoride-trifluoroethylene) [P(VDF-TrFE)] (PVDF) nanofibers (NFs) as the functional materials to detect human movements, physiological signals, the brightness of the light and volatile organic compounds [11].

An important theme in the field of smart clothes is the release of nanomaterials originally incorporated into a textile; it can occur by simple rubbing or contact with the skin, by sweating or during washing [12,13]. Sweating and continuous dermis friction can affect the degree of release of nanoparticles embedded in the fabric, increasing their dispersion in the human body. For this reason, the use of artificial sweat solution in toxicology evaluation studies is very widespread: in many in vitro studies, artificial sweat solution has been employed to assess the implications of skin exposition, testing the nanomaterial concentration released from fiber surfaces [14,15,16,17]. Since skin represents the main barrier against nanomaterials, several authors have assessed in vitro the toxic effects of these nanostructures on keratinocytes and/or fibroblasts [18,19,20,21].

There are also concerns about the release of nanomaterials to the environment as they may be accumulated in wastewater treatment plants or landfills, resulting in toxicity for aquatic and soil ecosystems [22,23,24,25].

This study is part of the project “Development of smart sensorized clothes for prevention and mitigation of risks for worker safety”, the main objective of which was to produce clothes made with engineered fabrics to monitor the physiological parameters of the worker. In particular, strain sensors, employed to measure the respiratory rate, are produced by depositing water-based inks suitably mixed with Graphene NanoPlatelets (GNPs) on fabric via the screen printing technique.

Herein, we evaluated the biocompatibility of engineered fabric with GNPs by means of in vitro and in vivo assays.

We investigated in vitro the biocompatibility on human keratinocytes (HaCaT) of engineered fabric using a cell viability test (MTT), morphological analysis by immunofluorescence and field emission scanning electron microscopy (FESEM). The FESEM was used to evaluate the morphological changes in the cell membrane as modifications in microvilli, such as elongation, bleb formation or reduction in their number, which are considered signals of cytotoxicity. Recent works report changes in the microvilli structure after exposing different cell lines to some nanomaterials [26,27,28,29].

In particular, we evaluated the effects induced on human keratinocytes by the eluates extracted from the smart fabric and by the contact of the engineered fabric with the cell line. In addition, the keratinocytes were seeded directly on the smart fabric to assess their adhesion, growth and morphology.

Moreover, we evaluated in vivo possible adverse effects of the graphene nanoplatelets eventually detached from the engineered fabric surface, using the model system *Caenorhabditis elegans*. Its easy handling, body transparency, small size, short life cycle and fully sequenced genome with a high human homology make *C. elegans* a very suitable model for nanotoxicology studies [30,31]. Several toxicity markers such as vitality, reproduction, larval development, motility and response to oxidative stress have been widely employed to investigate the negative consequences of different nanomaterials on this nematode [32].

## 2. Materials and Methods

### 2.1. Smart Fabric

The smart materials were developed from two different water-based commercial inks supplied by Amex and Eptaink, respectively. GNPs were produced starting from the thermal expansion of the Graphite Intercalated Compound (GIC), as described in Sarto et al. [33]. The nanomaterial was dispersed into the inks using a mixing process, and subsequently, the mixtures were deposited on fabrics (96% polyester and 4% elastane) using the screen printing method, as reported in Marra et al. [34]. Once the laying of the material on the fabric was finished, it was necessary to place the material in the oven for 3 min at a temperature of 150 °C in order to complete the polymerization (Figure 1).

The morphology of the fabric samples was studied using Zeiss Ultra Plus FESEM (8.0 kV; SE detector). The samples’ surface was coated with a 15 nm-thick Cr layer using a Quorum Tech Q150T sputter. The imaging of materials was produced at different magnifications.

### 2.2. Fabric Types Used

During the study, two different types of water-based inks were used, here referred to as polymer 1 (not containing formaldehyde) and polymer 2 (including a low amount of formaldehyde). Further details on ink composition are not available.

All the biocompatibility tests were performed on UnTreated (UT) fabric, considered as a control, on fabric treated with: Polymer 1 (P1), Polymer 2 (P2), Polymer 1 mixed with Graphene Nanoplatelets (P1GNPs) and Polymer 2 blended with Graphene Nanoplatelets (P2GNPs).

### 2.3. Eluates

To prepare the eluates, the different fabric types were first cut into small pieces (3 cm × 3 cm) and sterilized with UV light for 72 h. They were placed in 10 mL of Dulbecco’s Modified Eagle’s Medium (DMEM; Euroclone, Pero, MI, Italy) without fetal bovine serum (FBS; Sigma-Aldrich, MO, USA) in test tubes and stirred for 72 h at a constant speed of 240 rpm/min.

In the body text, abbreviations were used to indicate the eluate obtained from every single type of fabric, such as EUT, which refers to the UnTreated fabric, or EP1 and EP2 terms, for samples treated with Polymer 1 and Polymer 2, respectively. EP1GNPs indicate the resultant eluate derived from the fabric treated with Polymer 1 mixed with Graphene Nanoplatelets, while EP2GNPs refer to the one treated with Polymer 2 blended with Graphene Nanoplatelets.

### 2.4. In Vitro Assays

#### 2.4.1. HaCaT Cultures and Treatments

The human keratinocytes cell line HaCaT (kindly provided by Prof. Talora, Sapienza University of Rome) spontaneously immortalized from a primary culture of keratinocytes [35] was cultured in DMEM, supplemented with 10% FBS, 2 mM L-Glutamine and 100 units/mL penicillin and 100 mg/mL streptomycin (Sigma-Aldrich, St. Louis, MO, USA) at 37 °C with 5% CO_2_ in a humidified atmosphere.

Three types of treatments were performed to evaluate biocompatibility:➢HaCaT cells were exposed to the eluates obtained from different types of fabric;➢UV-sterilized small pieces of different types of fabric were placed directly in contact with HaCaT as reported in other studies [36,37];➢HaCaT cells were directly seeded on the different fabric types used.

#### 2.4.2. Cell Viability Assay

Cell viability was determined using the MTT assay as described by Bossù et al. [38]. Briefly, 3 × 10^4^ HaCaT cells were plated on 24-well culture plates and exposed or not to the fabric extracts at 8, 24, 48 and 72 h. Tetrazolium salts (MTT: 3-(4,5-dimethylthiazol-2-yl)-2,5-diphenyltetrazolium bromide (Sigma-Aldrich, St. Louis, MO, USA), 5 mg/mL suspended in PBS) were added to each well and incubated for 4 h. The formazan crystals were extracted from the cells with a solubilizing solution (DMSO). An ELISA reader (Multiskan™ FC Microplate Photometer, Thermo Fisher Scientific, Waltham, MA, USA) was used to measure the absorbance at a wavelength of 570 nm and reference length at 630 nm. The results were expressed as a percentage of the viability of untreated cells. The same experimental procedure was carried out by seeding 3 × 10^4^ HaCaT cells for a direct contact analysis for 4, 8 and 12 h or by seeding 5 × 10^4^ HaCaT cells directly on fabric pieces for 24, 48 and 72 h. Each experiment was performed three times in triplicate.

#### 2.4.3. Sample Preparation for FESEM and Analysis

HaCaT cells (5 × 10^5^), seeded on 13 mm diameter circular slides, were incubated with eluates obtained from the different fabric types for 48 h and placed in contact with small pieces of the different fabrics for 8 h. Unexposed cells were used as negative controls. Moreover, HaCaT cells (5 × 10^5^) were directly seeded on the different fabric types for 48 h. At least three independent experiments were performed per condition.

According to the standard procedure [39,40], samples were prepared for FESEM analysis. Treated and control samples were fixed with 2.5% glutaraldehyde (25% EM grade, Agar Scientific, Monterotondo, Italy) in Dulbecco’s phosphate-buffered saline (PBS) without calcium and magnesium (pH 7.4, Sigma-Aldrich, St. Louis, MO, USA) at room temperature for 30 min and rinsed 4 times in PBS. The OsO4 (Sigma-Aldrich, St. Louis, MO, USA) was applied at 1% in PBS for 1 h, and then cells were washed 4 times with PBS. The samples were dehydrated by a gradual passage with a series of alcohols with a progressively increasing concentration (20%, 50%, 70%, 95% and 100%), and finally, CO_2_ critical point drying was performed. The same protocol was used for different fabric types. After critical point drying, samples were mounted on aluminum stubs using double-sided conductive tape and sputtered with a thin layer of chromium (40 nm) by a high-resolution sputter coater (AGB7234 Agar).

Sample analysis was carried out with Ultra Plus FESEM (Zeiss, Jena, Germany). In this study, the samples were analyzed at a working distance of 2–3 mm, with an acceleration voltage of 5.0 kV using the InLens SE detector. For each test condition, three different independent experiments were performed, each in triplicate. The analysis was carried out by choosing fields in a stochastic way by setting the magnification to 5000× and systematically moving in a “Greek fret” way in order to fully observe each coverglass. For each sample, one hundred cells were randomly analyzed, and any morphological changes present on the surface of the cell membrane (such as microvilli density, enlargement, elongation and ruffling formation) were reported. The number of cells with the same variation was expressed as a percentage (%).

#### 2.4.4. Immunofluorescence Microscopy Analysis

For immunofluorescence analysis, 3 × 10^4^ HaCaT cells were seeded on glass coverslips and left to adhere. Then, cells were exposed to fabric extracts for 48 h and fixed with 4% paraformaldehyde for 30 min, followed by treatment with 0.1 M glycine in PBS for 30 min and with 0.1% Triton X-100 in PBS for an additional 5 min, to allow permeabilization. To analyze the actin cytoskeleton, HaCaT cells were incubated with TRITC-phalloidin (Sigma-Aldrich, St. Louis, MO, USA) (1:50) for 45 min at 25 °C. Nuclei were stained with 40,6-diamidino-2-phenylindole (DAPI) (Sigma-Aldrich, St. Louis, MO, USA). Coverslips were finally mounted with Mowiol (Sigma-Aldrich, St. Louis, MO, USA) for observation. The immunofluorescence signal of HaCaT cells was analyzed by recording stained images using an Axio Observer Z1 inverted microscope equipped with an ApoTome.2 System (Carl Zeiss Inc., Oberkochen, Germany). Digital images were acquired with the AxioCam MRm high-resolution digital camera (Zeiss, Jena, Germany) and processed with the AxioVision 4.8.2 software (Zeiss, Jena, Germany). ApoTome optical images of fluorescent HaCaT cells were recorded under a 40×/0.75 objective (Zeiss, Jena, Germany), using the Filter set 43, and an HBO mercury vapor lamp (Osram, Munich, Germany) was used as the excitation source.

### 2.5. In Vivo Assays

#### 2.5.1. Nematode Strains and Maintenance

The *C. elegans* strains used in this study are the N2 wild-type strain and the CL2166 (dvIs19 [(pAF15)gst-4p::GFP::NLS] III) as transgenic strain from *Caenorhabditis* Genetic Center (University of Minnesota, Minneapolis, MN, USA). All nematodes were grown in normal conditions at 16 °C on Nematode growth medium (NGM) agar plates seeded with OP50 *Escherichia coli* suspension as a feeding source [41].

#### 2.5.2. Sample Preparation for *C. elegans* Experiments

To simulate human sweating and skin contact, which combined could lead to a release of nanomaterials from engineered fabrics, sterilized texture samples (1.5 cm × 1.5 cm) covered by GNPs mixed with polymers (P1GNPs and P2GNPs) were soaked into sterile 5 mL of M9 (1×) saline solution (KH_2_PO_4_ 3g/L, Na_2_HPO_4_ 6 g/L, NaCl 5 g/L, MgSO_4_ 1 mL 1 M) for 72 h under continuous stirring. The final eluates were used to treat nematodes (EP1GNPs, EP2GNPs). Eluates were also obtained by washing polymer silk-screened fabric samples (EP1 and EP2) and uncovered ones (EUT Fabric).

#### 2.5.3. Survival Assay

About fifty one-day-adult animals derived from synchronized cultures grown on NGM-OP50 plates at 16 °C were transferred onto fresh NGM-OP50 agar plates and treated with 200 μL of eluates derived from washing fabrics (EP1, EP2, EP1GNPs, EP2GNPs and EUT Fabric) for the entire experiments. Furthermore, fifty untreated nematodes were fed continuously with *E. coli* OP50 on NGM agar plates and used as control (UT). In order to avoid interference of eluates on bacterial metabolism, the OP50 was supplied to nematodes as heat-killed. Every day animals were monitored, counted and moved to new plates with the addition of fresh eluate, thanks to the help of an Optech SZ-N series Stereomicroscope (Exacta + Optech, Modena, Italy). The nematodes were considered dead if they did not respond to a mechanical stimulus using a small platinum wire. All the experiments were made in duplicates and repeated at least three times.

#### 2.5.4. Brood Size Assays 

To assess possible toxicity effects on *C. elegans* reproduction, four fertile worms coming from NGMOP50 synchronous culture were transferred separately to NGM-OP50 plates and exposed to the various fabric eluates every day. The eggs were counted until the end of the reproductive period for each experimental condition.

#### 2.5.5. *C. elegans* Body Lengthiness

The probable impact of engineered fabrics release on nematodes’ development and growth was evaluated by measuring larvae body length treated or not with the different washing solutions starting from embryo hatching. Larvae were photographed at the indicated time points by using a Leica MZ10F stereomicroscope (Leica, Wetzlar, Germany) with a Jenoptik CCD camera (Jenoptik Jena, Germany), and their body measurements were determined using the Delta Sistemi IAS software (Delta Sistemi, Alessandria, Italy). An average of 40 nematodes was imaged in at least three independent experiments.

#### 2.5.6. Pharyngeal Pumping Analysis

The pharyngeal pumping rate of nematodes treated with different fabrics eluates was carried out, as described in [42]. This test was made in triplicate, and about 40 worms were considered in each experiment.

#### 2.5.7. Body Bend Analysis

Locomotion alterations were analyzed by measuring the body bend frequency of nematodes exposed to the different fabric washing solutions and comparing them to the control ones [43]. Specifically, nematodes were washed and placed in 10 μL of M9 buffer, allowing them to swim freely: body bending was evaluated on 20 animals for its condition during a period of 60 s.

#### 2.5.8. Fluorescence Microscopy Analysis for the Oxidative Stress Evaluation

One-day-adult transgenic CL2166 nematodes were subjected to diverse fabric eluates for 24 h and 48 h. After treatment, they were mounted onto 3% agarose pads containing 20 mM sodium azide and observed with a Axiovert 25 microscope (Zeiss, Jena, Germany) under a 32×/0.40 objective (Zeiss, Jena, Germany) using the Filter set 38HE, and an HBO mercury vapor lamp (Osram, Munich, Germany) was used as the excitation source. Quantification of fluorescence intensity was evaluated with ImageJ 1.43 (NIH) software measuring the ratio of pixels per area of worm. For each sample, 10 transgenic nematodes were analyzed, and the mean value was reported and expressed as median fluorescence intensity (MFI).

### 2.6. Statistical Analysis

All experiments were performed in triplicate. The statistical significance was determined by one-way or two-way ANOVA analysis coupled with a Bonferroni post-test (GraphPad Prism 5.0 software, GraphPad Software Inc., La Jolla, CA, USA) and defined as * *p* < 0.05, ** *p* < 0.01, and *** *p* < 0.001. Ns is not significant. One-way ANOVA analysis, followed by Student–Newman–Keuls post hoc analysis, was performed to compare the results of the FESEM analysis. Confidence levels of 0.05 and 0.001 were considered significant and highly significant, respectively.

## 3. Results

### 3.1. FESEM Morphological Analysis of Fabrics

The morphology of the fabrics has been analyzed by investigating the integration between matrix and filler, as well as the uniformity and homogeneity of the film, by FESEM. In the neutral fabric (Figure 2a), it is possible to distinguish the individual fibers of the yarn used to produce the fabric.

In Figure 2b,c, it is possible to compare the fabric samples with the two unfilled polymer coatings, evaluating the integration of the polymers on the surface and in the interstices of the fabric. Polymer 2 (Figure 2c) showed greater surface coverage than Polymer 1 (Figure 2b), with the single fibers completely submerged. This can be justified due to the lower viscosity of Polymer 2, which facilitates the wettability of the fabric. Figure 2d,e show fabric samples screen printed with the GNP-loaded polymers, where the difference in surface coverage persists, and this is due to the different viscosities of the polymers (P1GNPs and P2GNPs). The high-magnification images (Figure 2f,g) show the degree of surface coverage in both samples screen printed with GNP-based inks. However, the sample coated with Polymer 2 and GNPs (Figure 2g) shows greater homogeneity than Polymer 1 and GNPs (Figure 2f) with the nanometric filler. The surface of the screen-printed fabric in Figure 2f shows not covered areas and surface areas with filler on the top. Moreover, it is possible to observe the presence of aggregates, which are the cause and effect of a higher viscosity of Polymer 1.

### 3.2. Cell Viability

We evaluated the effect of fabric eluates obtained from UT, P1, P1GNP, P2 and P2GNP fabrics at 8, 24, 48 and 72 h. The results obtained and shown in Figure 3A indicate that the cells exposed to all types of fabric extracts at all times analyzed did not show significant changes compared to the HaCaT cells used as control, grown in the presence of the culture medium alone.

We also performed an MTT assay by growing HaCaT cells in 24-well plates and placed small pieces of the smart fabrics over the adherent cells for 4, 8 and 12 h to recreate direct contact between cells and fabric in vitro. As shown in Figure 3B, at 4 and 8 h of exposure to all the smart fabrics used—UT, P1, P1GNPs, P2 or P2GNPs—HaCaT cells did not show significant changes. On the other hand, HaCaT cells in direct contact with P1 and P1GNPs for 12 h showed a small reduction in vitality compared to the control sample (UT). Then, an additional set of MTT assays were performed to evaluate the biocompatibility of HaCaT cells grown directly on fabric. In this case, small pieces of sterilized UT, P1, P1GNP, P2 and P2GNP fabrics were placed inside a 24-well culture plate, on which the cells were seeded. The biocompatibility test was performed after 24, 48, 72 h and one week. The results shown in Figure 3C indicate that after 24 h, there were no significant differences between the various fabrics used. However, cells seeded on P2 and P2GNPs showed a higher rate of cell viability compared to P1 and P1GNPs (Figure 3C), which became more evident at 72 h and 1 week, suggesting a time-dependent response.

### 3.3. Morphological Analysis

#### 3.3.1. FESEM 

As previously mentioned, tests were also carried out on fabrics treated only with the two different polymers to assess their possible toxic effects.

FESEM observations of cells exposed to UT fabric showed that the cellular surface was covered by regular microvilli (Figure 4a). HaCaT cells exposed to the EP1 showed a microvilli accumulation on the cell surface in 13% of the cells (Figure 4b, white arrow) and a reduction in the number of the same in 5% (Figure 4b, black arrow). Conversely, the HaCaT cells treated with EP2 showed a microvilli accumulation in 5.5% of the cells and a reduction in the number of microvilli in 4.5%.

Five percent of the cells exposed to EP1GNPs displayed the presence on the cell surface of enlarged microvilli (Figure 4c) and a reduction in the number of microvilli in two percent of the cells. HaCaT treated with EP2GNPs showed the presence of elongated microvilli in 2% of the cells, and in some cells, this change was associated with a reduction in microvilli number (Figure 4d). All morphological changes found were significant (*p* < 0.05).

On the cell surface of the HaCaT on which the small pieces of the different types of fabrics were placed in contact for an exposure period of 8 h, no significant changes in the microvilli structure were found with respect to control cells in any sample analyzed, as shown in Figure 5a–d.

As concerning the cells seeded directly on the different types of fabrics, those grown on the UT fabric appeared elongated on the fabric fibers, completely covering them (Figure 6a,b). HaCaT cells grown on the P1 (Figure 6c) and P2 were both flat on the fabric and rounded. FESEM analysis showed that some cells grown on P2 fabric were coated by polymers (Figure 6d), which suggests that Polymer 2 without graphene nanoplatelets was less stable.

With regard to the P1GNP and P2GNP fabrics, they did not show significant alterations. P2GNP fabric was uniformly smooth, showing that GNPs stabilized Polymer 2, which no longer tended to melt. The cells grown on the P1GNP fabric were mostly spherical in shape, but in a few places, the cells were completely spread out, covering the fabric. Furthermore, cells tended to localize in fabric cavities where the polymer distribution was not uniform (Figure 6e). Cells grown on P2GNPs were spread out and completely covered the fabric (Figure 6f). 

#### 3.3.2. Immunofluorescence Analysis

To evaluate the organization of the actin cytoskeleton in HaCaT cells, a morphological analysis was performed by immunofluorescence microscopy. To this aim, human keratinocytes were exposed to UT, P1GNP and P2GNP fabric extracts for 48 h and stained with TRITC-phalloidin, which specifically recognizes filamentous actin cytoskeleton. The morphological analysis shows that the cells grown in the presence of all fabric extracts exhibited the typical polygonal-shaped morphology of HaCaT cells, with actin filaments mainly organized in stress fibers, highly similar to the control cells treated only with the culture medium (Figure 7). These results suggest that the fabric extracts did not induce any morphological changes in the cells.

### 3.4. In Vivo Results

The *C. elegans* model was employed to assess in vivo the toxicity of the different fabric eluates, which could release GNPs during the washing phase or sweating when we are considering their possible applications. In recent toxicology studies of engineered fabrics, artificial sweat is used as a dispersion liquid to test the release of nanomaterials. The excessive acidity of artificial sweat is not compatible with *C. elegans* in vivo tests, so the GNP liquid dispersion was simulated using a saline buffer currently employed in nematode analysis (M9 buffer). As the first screening of eluates for in vivo effects, *C. elegans’* lifespan was analyzed. To this aim, age-synchronized young adult worms were continuously exposed to EUT, EP1, EP2, EP1GNP and EP2GNP aliquots, prepared as previously described. There were not observed any considerable changes in the survival of treated nematodes compared with the untreated ones (UT) used as controls (Figure 8A).

Later, possible genotoxic effects on treated animals’ progeny were assessed through the brood size analysis, which consists of counting the number of embryos produced per single worm. According to this, wild-type hermaphrodites were exposed to eluate samples from the L4 stage to the end of the fertility period, and the number of embryos laid was considered. As shown in Figure 8B, no significant differences in the number of eggs laid per worm were identified for all types of eluates when compared to the untreated nematodes (UT). Meanwhile, potential delays in the larval development of treated worms were investigated by measuring their body length, from embryonic to adult stages. Figure 8C shows no developmental differences in the animals subjected to eluates during their whole growth compared to the untreated ones.

Furthermore, potential damages to nematodes’ neuromuscular apparatus were analyzed by evaluating parameters such as the pumping-rate analysis (Figure 8D), which consists of counting how many times per minute the contraction of the grinder takes place and the body bending examination (how many times in a minute nematodes are able to make a complete movement of the body with a dorsal–ventral curvature that spreads from the head to the tail) (Figure 8E) [44]. The grinder is located in the terminal bulb of the *C. elegans* pharynx, and its function is to grind ingested bacteria, allowing their passage to the intestinal lumen. Moreover, no significant differences were found, demonstrating the non-toxicity of the treated fabrics made for this purpose. Therefore, the assessment of all the phenotypes reported in Figure 8 revealed the absence of discrepancies in the survival, in the larval development, as well as in the fertility of animals treated with eluates derived from fabrics with nanomaterials (EP1GNPs, EP2GNPs) and eluates relative to fabrics coated with only polymers and not coated (EP1, EP2, EUT). Moreover, graphs trends appeared to be in line and fully comparable to those of control nematodes, the untreated ones (UT).

In addition, the oxidative stress of nematodes exposed to the different eluates was taken into account, thanks to the evaluation of the expression of the gst4 gene encoding for the glutathione-S-transferase. Specifically, the transgenic strain of *C. elegans* was exploited: in this mutant, gst-4 was fused to the GFP (Green Fluorescent Protein) reporter gene in order to observe its expression in the animal bodies by fluorescence microscopy analysis. This enzyme is involved in the mechanisms of abatement of reactive oxygen species produced in *C. elegans* innate immunity during infections or in xenobiotic detoxification [45,46].

The fluorescence evaluation was carried out at 24 h and 48 h of nematodes exposed to the various eluates plated (Figure 9a,b). In the transgenic strain analysis, an oxidative stress induction effect was not found. In fact, in the graphs shown in Figure 9c,d, the fluorescence intensities of the all animals (*C. elegans* CL2166 strain) treated were similar to those of untreated ones and remained unchanged over the hours.

## 4. Discussion and Conclusions

The biological evaluation of smart fabrics or wearable devices starts from the determination of biocompatibility and cytotoxicity. Human dermal fibroblast and epidermal keratinocytes are the cell lines commonly used for testing skin compatibility. Several authors exposed human keratinocytes to different graphene-based materials. Fusco et al. exposed HaCaT cells to low-concentrations of few-layer graphene (FLG) and dehydrated graphene oxide (d-GO), analyzing cell viability (WST-8) and TEM analysis. The results showed that these materials did not affect cell viability, and both FLG and d-GO were taken up by HaCaT cells [47]. Li et al. investigated the interaction of graphene and few-layer graphene microsheets with three cell lines: primary human keratinocytes, human lung epithelial cells and murine macrophages. Bioimaging experiments performed with confocal fluorescence microscopy and electron microscopy (TEM and SEM) showed graphene internalization, and the authors demonstrated by SEM observations that this occurred by membrane penetration of the graphene microsheet corner or edge [20].

In this study, we checked the possible release of GNPs from the smart fabrics, evaluating in vitro the cytotoxicity of HaCaT cells exposed to different extracts of fabrics covered or not with GNPs. The absence of significant changes in the treated cells suggests the non-release of GNPs or the release in very small concentrations from the textiles used, a result in agreement with that of Salesa et al., who reported that low concentrations of GNPs were not toxic for HaCaT cells [48].

Moreover, we confirmed these data by fluorescence analysis, which shows cells in good health, further suggesting that the extracts obtained from the different smart fabrics used in this study do not contain toxic material for the cells.

In vitro human keratinocytes grown in direct contact with smart fabrics covered or not with GNPs, show a high rate of viability, suggesting that these fabrics are unable to induce inflammatory or hypersensitivity reactions on human skin. A similar result was obtained by Stan et al., which demonstrated the absence of cytotoxicity of dermal fibroblast cells after short-term exposure to Cotton Knit Coated with Fe-N-Doped Titanium Dioxide Nanoparticles [37]. Moreover, our results show that HaCaT cells grown directly on different fabrics showed a higher viability rate for cells grown on P2 and P2GNPs compared to P1 and P1GNPs after a long time. This result could be correlated with different distribution of P1 and P2 as demonstrated by morphological characterization of different samples. Indeed, P2, unlike P1, covers a higher tissue surface uniformly, probably creating a larger surface, which could promote the adhesion of the cells to the substrate.

In the literature, many in vitro biocompatibility studies [36,37,49,50,51] are reported, but only a few studies use scanning electron microscopy to evaluate changes in cell morphology [52,53]. In this work, cell membrane changes in keratinocytes following incubation with eluates or after direct contact with different types of fabric were analyzed by FESEM, as changes in the structure of the microvilli are indicators of cytotoxic effects. Moreover, cell shape alterations were assessed on HaCat cells grown on fabrics. The functions performed by microvilli on the cell surface are involved in almost all fields of cell physiology such as volume regulation, light perception, Ca^2+^ signaling, etc. [54]. A role of microvilli, through their generation or elongation, is to keep the cell surface free of cytotoxins [54]. The internalization of graphene oxide (GO) and carboxyl graphene nanoplatelets were observed in the human hepatocellular carcinoma cell line (Hep G2), non-phagocytotic cells [52]. Hep G2 cell surface was completely or partially covered with graphene nanoplatelets depending on the concentration used and interactions between microvilli and micro-sized platelets were observed. Our FESEM observations never displayed GNPs deposited on the cell surface and/or interaction between microvilli and GNPs; however, changes on the cell membrane level were observed after incubation with eluates. The main alterations were the reduction in the number of microvilli and changes in their structure. Similar alterations were observed in A549 exposed to MWCNTs-COOH [29] or to TiO2 nanoparticles [28]; in the human gastric carcinoma cell line exposed to chitosan nanoparticles [26]; in the human intestinal cell line (Caco-2) after exposure to titanium dioxide (TiO2) [55]; in HeLa cells treated with magnetic iron oxide nanoparticles (MIONPs) [56] or exposed to static magnetic fields [57].

No significant morphological alteration was detected in the cells placed in contact with smart fabrics compared to control. 

HaCaT cells grown on P2GNPs were spread with typical actin protrusions, and they covered the textile, while cells grown on P1GNPs exhibited a round shape. Slepička et al., to enhance the cytocompatibility of a perfluoroethylenepropylene (FEP) foil, treated it with different plasma power. HaCaT cells grown on pristine FEP exhibited a round shape and limited intracellular connections in comparison with cells seeded on plasma-treated FEP foil that was spread out [53].

The experiments performed with fabric treated with P1GNPs demonstrated the change presence at cell surface level after exposure to eluates and a large number of spherical-shaped keratinocytes on the cells grown directly on fabric. These results are in agreement with those of the MTT test related to the growth of cells directly on the fabric, suggesting that the spherical cells observed at FESEM, although still viable, detach from the fabric, as shown by the reduction in cell viability with increasing incubation times. Wan et al., in a recent study on the incorporation of double nanoplatelets into a natural polymer to make nacre-like nanocomposites, also used differences in cell morphology (cell spreading area) to evaluate the biocompatibility [58].

The nematode *C. elegans* has become an important in vivo alternative assay system to evaluate the safety of nanomaterial applications, especially in initial biological screenings of new nanoparticles. This model organism has been successfully applied to assess the potential risks to human and environmental health, thanks to its numerous advantages of use [31]. Even if these animals do not have any specific organs such as eyes, heart or kidneys, they can be suitable to understand the interaction between nanomaterials and biological barriers similar to those of mammals, such as the skin and intestine [59]. In this work, we focused on *C. elegans* sublethal effects eventually induced by GNPs released from fabrics, including the survival, reproduction, development, neuro-muscular functions, as well as oxidative stress. Previous studies identified ROS production as the main toxicity mechanism of graphene nanomaterials in *C. elegans*, especially for graphene-oxide, which can activate the antioxidant response during continuous exposure [60,61,62,63]. Despite this, our analysis carried out on *C. elegans* GST4:GFP transgenic mutant did not show any differences at the fluorescence intensity level compared to the control. Although it has been reported that graphene-based nanomaterials induced harmful effects in different model systems, including also *C. elegans*, none of the investigated endpoints in this study highlighted the GNP toxicity [64,65]. All the results were confirmed by our previous works, where these graphene nanosheets in suspension did not affect *C. elegans* lifespan and reproductive ability [66,67,68]. Salesa et al. exposed *C. elegans* to a wide range of GNP concentrations for acute and chronic toxicity determination, demonstrating their no significant toxicity at low amounts. Indeed, GNPs can be administered to the animals at the non-toxic concentrations of 25 and 12.5 µg/mL for 24 h and at 12.5 µg/mL for 72 h of exposure [48]. According to this, GNP toxicity resulted in dosage-dependent exposure. Here, the lack of oxidative stress and noxious effects in *C. elegans* was probably due to the absence or the extremely low concentration of dissolved GNPs into the fabric eluates. The biocompatibility of GNPs was also verified in nematodes treated with these nanoplatelets when they were decorated with zinc oxide nanorods (ZNGs). Even in this case, no negative outcomes were observed in the physiology of worms treated with ZNGs [69]. *C. elegans* was also used to evaluate the biocompatibility and the distribution of halloysite nanotubes attractive as carriers for new antimicrobial agents [70].

Data obtained from the *C. elegans* model and keratinocytes were helpful to determine the non-toxic properties of the smart fabric, ensuring it’s environmental and health safety. Indeed, the results of the cell viability test and FESEM analysis of the smart fabric with P2 agreed with the data obtained from *C. elegans*, while the smart fabric with P1 exhibited slight cytotoxicity.

These results provide a direction for future applications of graphene-based elastic fabric as a biocompatible piezoresistive sensor.

## Figures and Tables

**Figure 1 nanomaterials-12-01405-f001:**
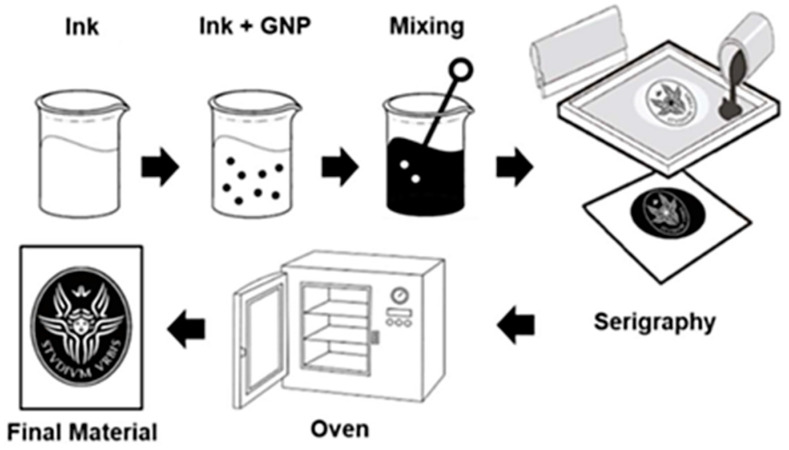
Production Process.

**Figure 2 nanomaterials-12-01405-f002:**
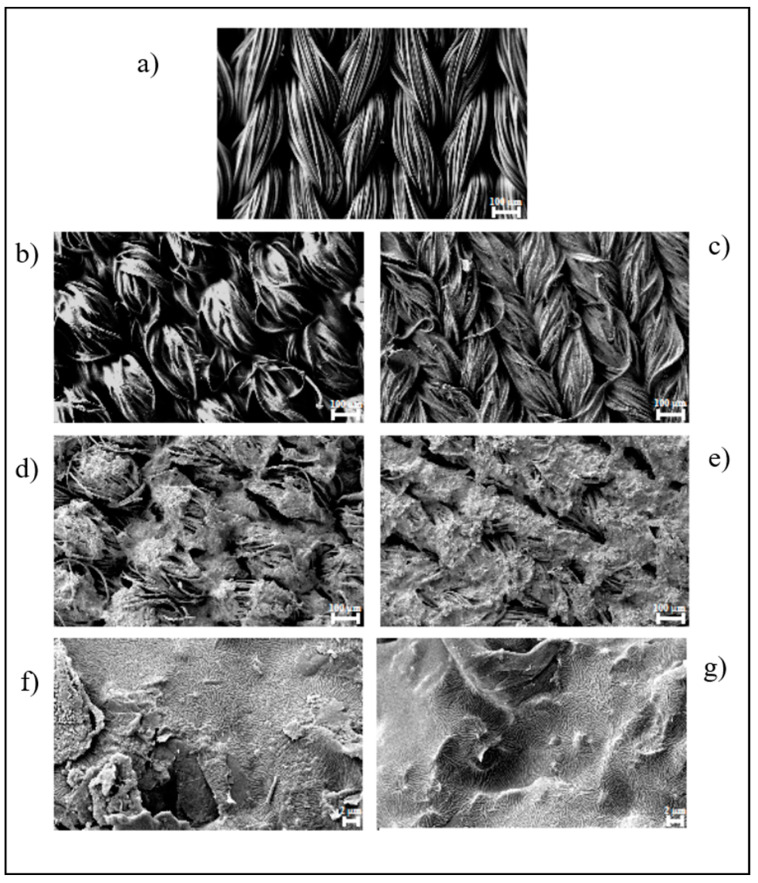
FESEM images of: (**a**) neutral fabric UT (Bar 100 µm); (**b**) P1 screen printed fabric (Bar 100 µm); (**c**) P2 screen printed fabric (Bar 100 µm); (**d**,**e**) screen printed fabric with P1GNPs; (**f**,**g**) screen printed fabric with P2GNPs at different magnifications (Bar 100 µm and Bar 2 µm, respectively).

**Figure 3 nanomaterials-12-01405-f003:**
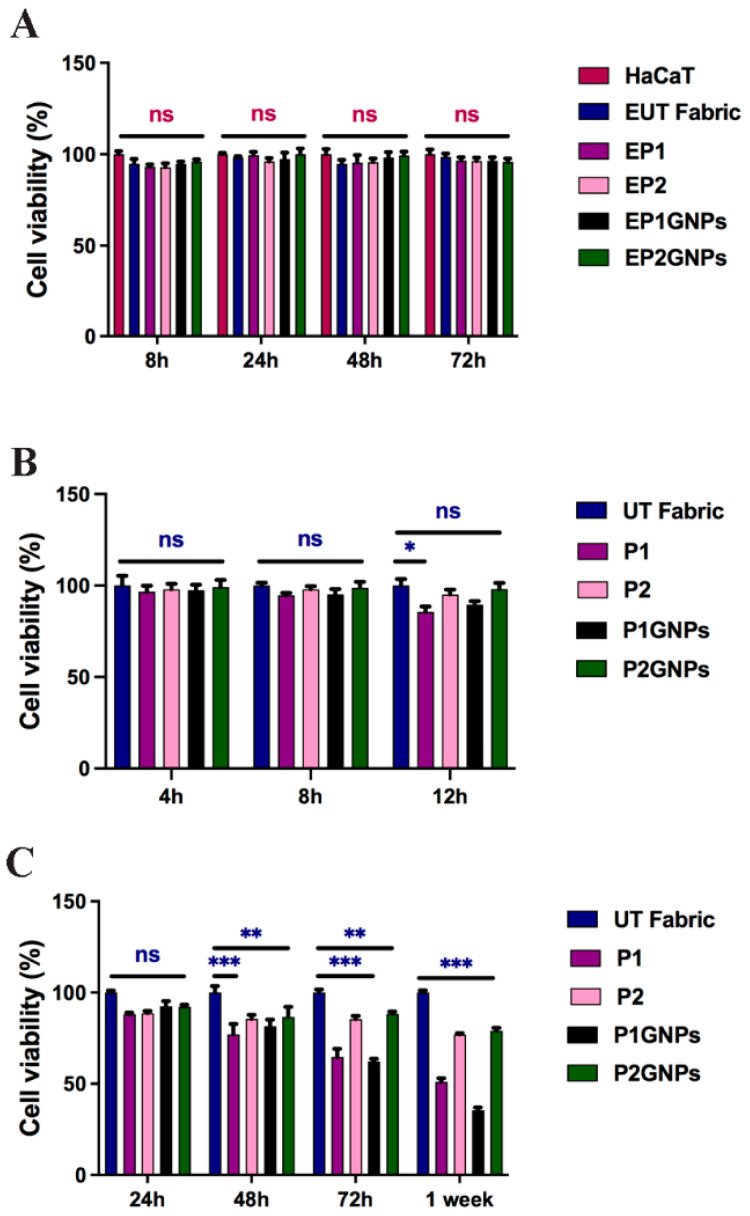
Cell viability evaluated by MTT assay of HaCaT cells grown: (**A**) in the presence of fabric extracts at 8, 24, 48 and 72 h; (**B**) in direct contact with smart fabrics at 4, 8 and 12 h; (**C**) directly on smart fabrics for 24, 48, 72 h and 1 week. The data are normalized to: (**A**) untreated HaCaT cells; (**B**) HaCaT cells grown in direct contact with untreated fabric; (**C**) HaCaT cells grown on untreated fabric (UT) and reported as a percentage. Statistical analysis was performed by the two-way analysis of variance (ANOVA) method coupled with the Bonferroni post-test. (* *p* < 0.05; ** *p* < 0.01; *** *p* < 0.001; ns: not significant).

**Figure 4 nanomaterials-12-01405-f004:**
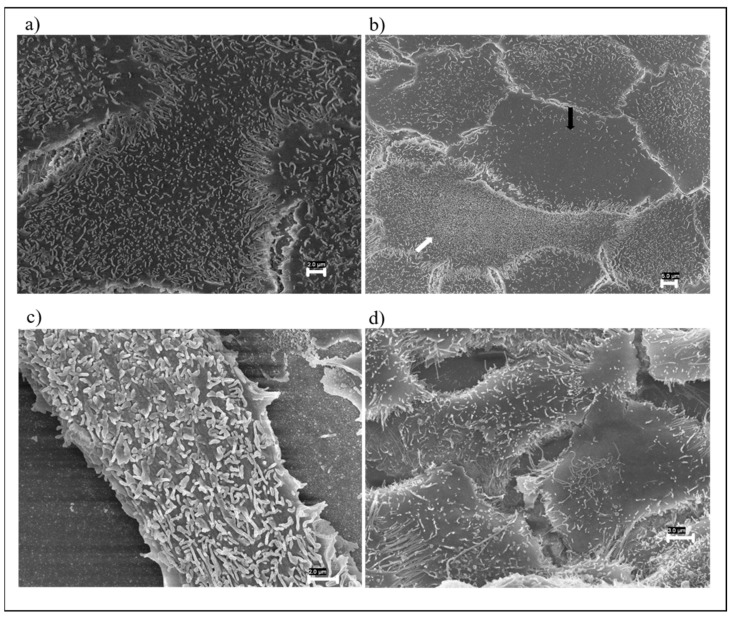
FESEM images of HaCaT cells exposed to: (**a**) EUT (Bar 2 µm) and (**b**) EP1 (Bar 5 µm), where white and black arrows indicate microvilli accumulation and a reduction in the number of microvilli on the cell surface, respectively; (**c**) EP1GNPs (Bar 2 µm), which displays the presence of enlarged microvilli on cell membrane; (**d**) EP2GNPs (Bar 3 µm). HaCaT cells show elongated microvilli.

**Figure 5 nanomaterials-12-01405-f005:**
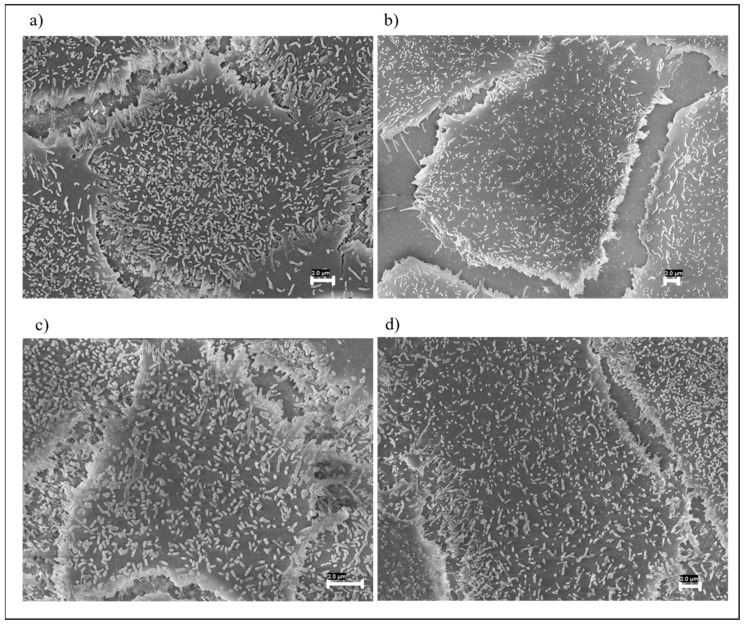
(**a**) FESEM images of control cells (Bar 2 µm). Cells in contact with different types of fabric: (**b**) UT (Bar 2 µm); (**c**) P1GNPs (Bar 2 µm); (**d**) P2GNPs (Bar 2 µm).

**Figure 6 nanomaterials-12-01405-f006:**
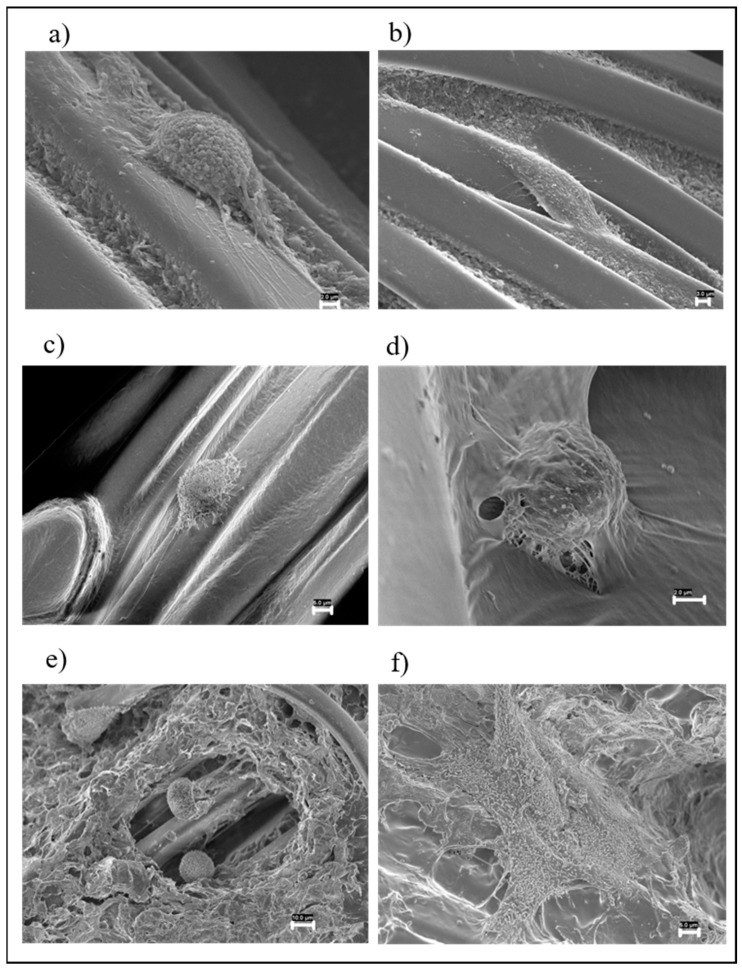
HaCaT cells grown on fabrics: (**a**) (Bar 2 µm) and (**b**) (Bar 3 µm) UT; (**c**) P1 (Bar 5 µm); (**d**) P2 (Bar 2 µm); (**e**) P1GNPs (Bar 10 µm); (**f**) P2GNPs (Bar 5 µm).

**Figure 7 nanomaterials-12-01405-f007:**
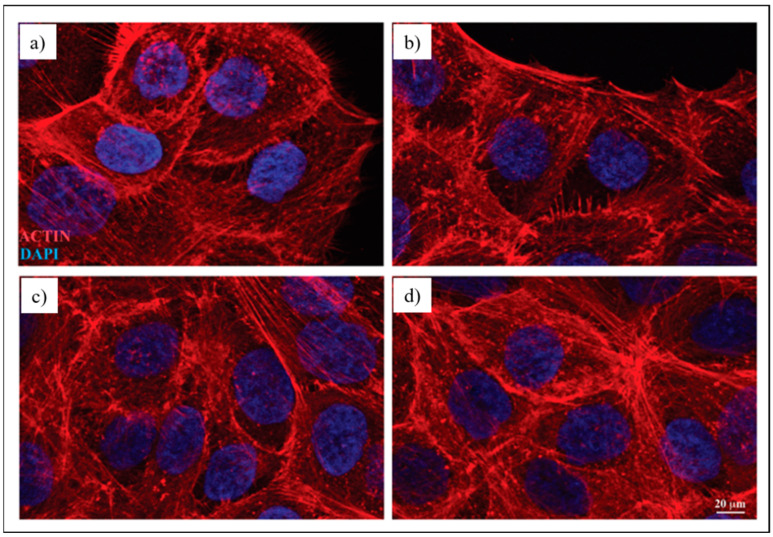
Immunofluorescence images of HaCaT cell morphology (**a**) without exposure or after exposure to: (**b**) EUT, (**c**) EP1GNP or (**d**) EP2GNP fabric extracts for 48 h. The actin cytoskeleton is highlighted by staining with TRITC-phalloidin, and the nucleus is detected by DAPI. Bar: 20 µm.

**Figure 8 nanomaterials-12-01405-f008:**
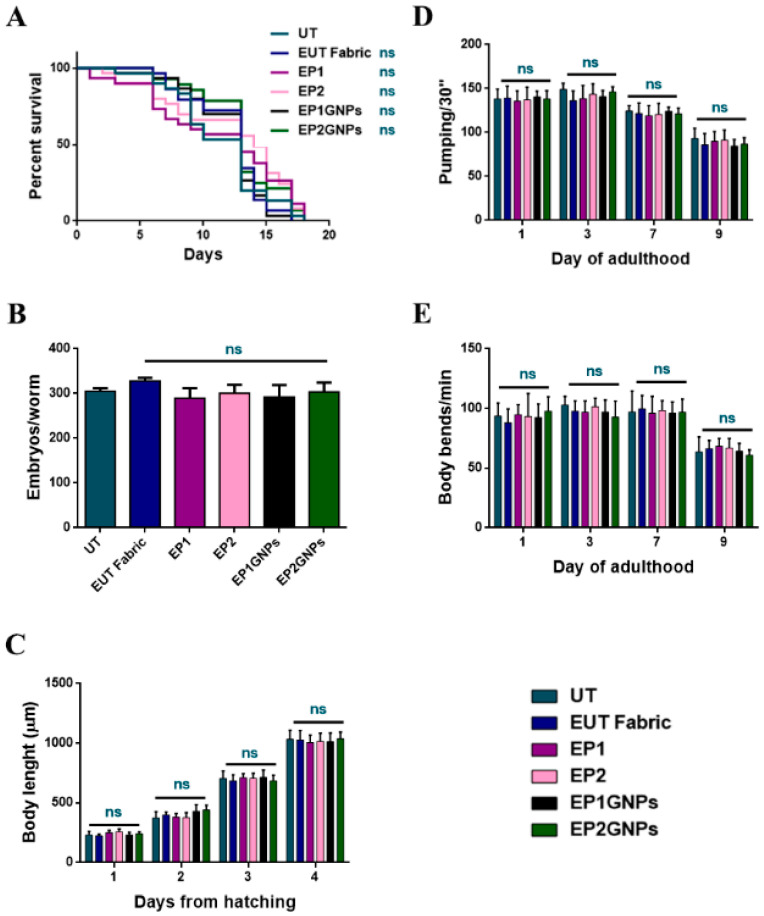
(**A**) Survival curve of young adult N2 worms untreated (UT) or treated, in continuous, with the indicated eluates. (**B**) The ability of *C. elegans* to produce offspring after extracts exposition. (**C**) Measurement of worm’s body length starting from their hatching on plates seeded with the indicated eluates. Worm length was measured from head to tail at the indicated time points. (**D**) Pharyngeal pumping rate after continued exposure to the indicated texture eluates and untreated worms used as control. The contractions were measured for 30 s and reported as mean pumping frequency for each condition. (**E**) Body bends frequency of nematodes treated with texture solutions from the one-day-adult stage and compared to untreated ones. The number of thrash was measured in a time period of 60 s. Statistical analysis was evaluated by one-way or two-way ANOVA with the Bonferroni post-test (ns as non-significant). Bars represent the standard deviations.

**Figure 9 nanomaterials-12-01405-f009:**
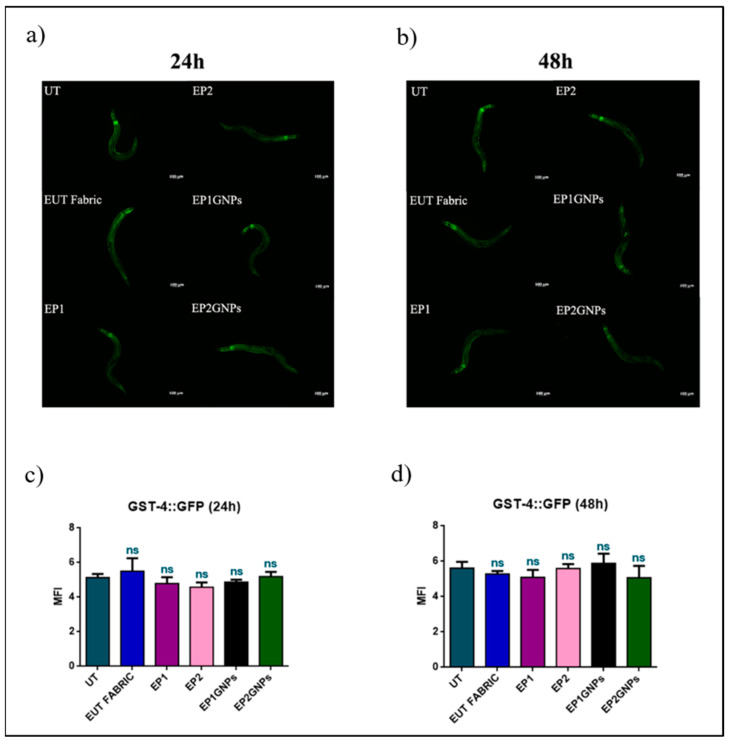
Fluorescence microscopy of GST4: GFP nematode strain after (**a**) 24 h and (**b**) 48 h of eluates treatment. Scale bar = 100 μm. Quantification of fluorescence intensity of *C. elegans* GST4::GFP transgenic strain exposed to the different texture extracts for (**c**) 24 h and (**d**) 48 h. Mean fluorescence intensity (MFI) was obtained using ImageJ software, and data were expressed as (mean ± SD), considering at least 10 worms in each experimental group (ns: not significant).

## Data Availability

The published data are all included in this manuscript.

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
