# Peer review of "In Vitro and In Vivo Biocompatibility Studies on Engineered Fabric with Graphene Nanoplatelets"

_nanomaterials, 2022, doi:10.3390/nano12091405_

Round 1
Reviewer 1 Report
The paper reports the preparation of the strain sensors by depositing two different types of water-based inks and mixed with graphene nanoplatelets on a fabric. The biocompatibility, the cell viability, themorphological analysis and caenorhabditis elegans tests of GNP fabric have been carried out in vitro and in vivo. This manuscript offers biocompatibility studies on an engineered fabric with graphene nanoplatelets, and I recommend acceptance of this manuscript after the following minor issues are addressed.
The GNP is attached to the fiber surface by physical adsorption. The authors should provide the stability information of the GNP on fabrics. Does it easy to fall off fabrics?
Author Response
The paper reports the preparation of the strain sensors by depositing two different types of water-based inks and mixed with graphene nanoplatelets on a fabric. The biocompatibility, the cell viability, themorphological analysis and caenorhabditis elegans tests of GNP fabric have been carried out in vitro and in vivo. This manuscript offers biocompatibility studies on an engineered fabric with graphene nanoplatelets, and I recommend acceptance of this manuscript after the following minor issues are addressed.
The GNP is attached to the fiber surface by physical adsorption. The authors should provide the stability information of the GNP on fabrics. Does it easy to fall off fabrics?
The production of the loaded ink takes place by dispersing the GNPs inside the screen printing ink following the scheme in figure 1. The commercial inks chosen can be mixed with glitter about 100 microns in size. GNPs have lateral dimensions ranging from 0.5 to 25 micron,and thickness up to 10-20 nm, so there were no problems of dispersion of the nanomaterial inside the polymeric matrix, which was subsequently deposited by screen printing on the fabric and polymerized in an oven at 150°C.
As reported in figure 2 (F and G) SEM analysis of printed fabric show the trapping of GNPs inside the fabric through the polymeric action. There is therefore a physicochemical adhesion of the ink on the fabric and an incorporation, by means of chemical polymerization, of the nanoplachets inside the ink.
Moreover, observations with SEM microscope of cells treated with the eluates (Figure 4) did not reveal the presence of GNPs, strongly supporting their stability inside the fabric.

Reviewer 2 Report
This paper reports interesting and convincing results, I suggest its publication following some revisions.
1) Which objective, fluorescence filter cube and excitation source was used for optical microscopy imaging? Same applies to "2.5.8. Fluorescence microscopy analysis for the oxidative stress evaluation"
2) Please provide good quality scale bars in all SEM images reported
3) Did the authors consider the effects of eluates on microbial communities in the nematodes? Can this be commented (see also the following question)?
4) Previously, some studies have demonstrated the distribution of nanoparticles in the nematodes intestines. Did the authors investigate the distribution? If no, it will be a good idea to cite the papers reporting the similar evaluation, as the following:
https://doi.org/10.1021/acsami.9b07499
Technical comments:
Figure 1 is too blurry, can it be replaced with a better quality image?
"in Lens detection" - perhaps, the InLens Zeiss TM mode is meant here?
C. elegans should be given in italics everywhere in the text
Line 240 lengthiness - is "length" meant here?
Author Response
This paper reports interesting and convincing results, I suggest its publication following some revisions.
1) Which objective, fluorescence filter cube and excitation source was used for optical microscopy imaging? Same applies to "2.5.8. Fluorescence microscopy analysis for the oxidative stress evaluation"
The requested information has been added to materials and methods section, at paragraph 2.4.4 and 2.5.8. respectively.
2) Please provide good quality scale bars in all SEM images reported
We improved the scale bar quality of all SEM images.
3) Did the authors consider the effects of eluates on microbial communities in the nematodes? Can this be commented (see also the following question)?
We did not consider the effects of eluates on microbial communities since to avoid the interference of bacterial metabolism on the evaluation of toxicity, the E.coli strain was supplemented to nematodes heat killed. A sentence has been added to the materials and methods section, we apologize for the lack of clarity.
4) Previously, some studies have demonstrated the distribution of nanoparticles in the nematodes intestines. Did the authors investigate the distribution? If no, it will be a good idea to cite the papers reporting the similar evaluation, as the following:
https://doi.org/10.1021/acsami.9b07499
We did not investigate the distribution of GNPs in the nematodes due to the lack or the very low amount of nanomaterial in the eluates. The reference has been added in the discussion section.
Technical comments:
Figure 1 is too blurry, can it be replaced with a better quality image?
We replaced Figure1 with a better quality image.
"in Lens detection" - perhaps, the InLens Zeiss TM mode is meant here?
We changed "in Lens detection" with “using InLens SE detector.
- elegans should be given in italics everywhere in the text
The modifications have been made
Line 240 lengthiness - is "length" meant here?
Yes, it means length
